# Experiential Learning Interventions and Healthy Eating Outcomes in Children: A Systematic Literature Review

**DOI:** 10.3390/ijerph182010824

**Published:** 2021-10-15

**Authors:** Sumantla D. Varman, Dylan P. Cliff, Rachel A. Jones, Megan L. Hammersley, Zhiguang Zhang, Karen Charlton, Bridget Kelly

**Affiliations:** 1Early Start, University of Wollongong, Wollongong, NSW 2522, Australia; dylanc@uow.edu.au (D.P.C.); rachelj@uow.edu.au (R.A.J.); mhammers@uow.edu.au (M.L.H.); zhiguan1@ualberta.ca (Z.Z.); bkelly@uow.edu.au (B.K.); 2School of Health & Society, Faculty of Arts, Social Sciences and Humanities, University of Wollongong, Wollongong, NSW 2522, Australia; 3School of Education, Faculty of Arts, Social Sciences and Humanities, University of Wollongong, Wollongong, NSW 2522, Australia; 4Illawarra Health and Medical Research Institute, Wollongong, NSW 2522, Australia; karenc@uow.edu.au; 5Faculty of Kinesiology, Sport and Recreation, University of Alberta, Edmonton, AB T6G 2H9, Canada; 6Faculty of Science, School of Medicine, Medicine and Health, University of Wollongong, Wollongong, NSW 2522, Australia

**Keywords:** healthy eating, nutrition, children, preschool, primary school, intervention, systematic review

## Abstract

Experiential learning is the process where learners create meaning from direct experience. This systematic review aimed to examine the effects of experiential learning activities on dietary outcomes (knowledge, attitudes, behaviors) in children. Four databases: Education Research Complete, Scopus, Web of Science and PsychINFO were searched from database inception to 2020. Eligible studies included children 0–12 years, assessed effect of experiential learning on outcomes of interest compared to non-experiential learning and were open to any setting. The quality of studies was assessed using the revised Cochrane risk of bias tool by two independent reviewers and effect size was calculated on each outcome. Nineteen studies were conducted in primary school, six in pre-school and one in an outside-of-school setting and used nine types of experiential learning strategies. Cooking, taste-testing, games, role-playing, and gardening were effective in improving nutrition outcomes in primary school children. Sensory evaluation, games, creative arts, and storybooks were effective for preschool children. Multiple strategies involving parents, and short/intense strategies are useful for intervention success. Experiential learning is a useful strategy to improve children’s knowledge, attitudes, and behaviors towards healthy eating. Fewer studies in pre-school and outside of school settings and high risk of bias may limit the generalizability and strength of the findings.

## 1. Introduction

Overweight and obesity among children is a global public health concern [1,2]. Excess weight in childhood is associated with an increased risk of developing physical, social, and psychological conditions, overweight and obesity and earlier onset of non-communicable diseases [3,4], such as diabetes [5] and cardiovascular disease [6]. Healthy eating is essential in the early years of life (0–12 years) to ensure optimal growth and development, as well as reducing the risks of life-long health problems [7]. Data from several surveys of fruit and vegetable intake of children conducted globally [8,9,10], ref. [11] have reported low intakes of fruit and vegetables in children of between two to three portions compared to the recommended five portions per day [12]. Given children’s low adherence to nutrition recommendations, interventions that target nutritional knowledge, attitudes and dietary behaviors are needed. Nutrition-related knowledge [13], attitudes [14] and eating behaviors [15,16] learned in childhood have been shown to track into later years; therefore, it is imperative to establish healthy eating behaviors early in life [17]. The World Cancer Research Fund’s Nourishing Framework has provided a repository of global policy actions that promote healthy diets and reduce obesity and identifies behavior change communication as a key policy area [18]. Experiential learning approaches such as gardening and cooking may be more engaging to children compared to more traditional learning approaches, in which children are more passive recipients of the information [19,20]. Positive behaviours, attitudes, and knowledge of healthy eating in children have been successfully demonstrated through using experiential learning approaches [21,22,23]. Experiential learning is beneficial because it exposes children to hands-on experiences and active engagement with activities promoting critical thinking [24]. Experiential learning-based approaches can be a useful strategy to improve children’s knowledge, attitudes, and behaviors towards healthy eating because they enable children to experiment, explore, play, and become familiar with materials and concepts that are related to the targeted behaviours [25,26].

In the case of improving children’s knowledge, attitudes, and behaviors towards healthy eating, experiential activities such as taste-testing, cooking, gardening, games, and role-play can actively involve children with hands-on experiences, engagement with activities and extend their thinking and curiosity [24,27,28,29]. It may be a particularly useful strategy for pre-school and community settings as these are ideal settings to develop personal understanding, knowledge, skills, and attitudes through active engagement and reflection on their experiences and activities.

Two previous reviews on primary schools have found experiential learning approaches to be effective in improving nutrition-related behaviors, knowledge, and attitudes in children. A systematic review by Dudley et al. [22] also found that school-based interventions that were inclusive of experiential learning strategies, such as cooking, preparing food or gardening were associated with the largest effects in increasing nutritional knowledge, preferences and consumption of fruit and vegetables and reducing energy intake in primary school children compared to interventions without experiential learning components. Another systematic review reported on characteristics of successful nutrition-related experiential learning interventions and found that cooking-related activities and gardening increased children’s willingness to taste unfamiliar foods (e.g., new fruits and vegetables) and increased nutritional knowledge in primary school children [30]. Both reviews provide important contributions to the field; however, these reviews did not include children below five years of age and only focused on the primary/elementary school setting. Additionally, because the review by Charlton and colleagues (2020) aimed to identify the key characteristics of successful nutrition-focused experiential learning interventions for children, only effective interventions were included. Consequently, unbiased estimates of intervention effects (i.e., by comparing effective and ineffective interventions that used the same approach) could not be provided. To extend the existing knowledge in this area, this systematic review aimed to examine the effects of experiential learning activities among a broader age range of children (0–12 years), and in a broader range of settings including both school/pre-school and community settings, to provide a more comprehensive assessment of experiential learning opportunities for children.

## 2. Methods

This review was registered with PROSPERO International prospective register of systematic reviews (no. CRD42018103629) and adheres to the PRISMA (Preferred Reporting Items for Systematic Review and Meta-Analysis) statement for systematic reviews [31,32] to ensure transparent reporting.

### 2.1. Search Strategy

Four databases were searched for eligible studies: Education Research Complete, Scopus, Web of Science and Psych Info. Search terms used to obtain relevant studies were guided by the PICO approach: Population children between birth to age 12 years old (Children 0–12 years); Intervention (experiential learning activities); Comparison (no or non-experiential learning activity); Outcome behaviors i.e., food intake, knowledge, attitudes). Reference lists of included studies were also hand searched. The search terms are shown in the table below (Table 1). No limits were applied to the publication date and the search was conducted (“to obtain articles published from database inception to 2020”).

### 2.2. Eligibility Criteria

Included studies were randomized controlled trials (RCTs) or cluster/group RCTs (CRCTs) reported in original, peer-reviewed articles. Studies were excluded if they were not published in English, were non-experimental designs or were reviews or opinion articles and were not an outcome of interest. A post hoc protocol deviation was made to exclude non-randomized controlled trials and non-controlled trials because of the higher than anticipated number of RCTs and CRCTs identified. Only RCTs and CRCTs were included as these were deemed to be the most robust level of evidence. Eligible studies were identified using the Population, Intervention, Comparison, Out-comes (PICO) framework.

*Population:* For the purpose of this review, children were defined as those aged between birth to age 12 years old (0–12 years). Studies were included if the mean age of participants was between 0–12 years and excluded if the mean age of participants was above 12 years at baseline of the intervention. This age range was selected for this review because children 0–5 years were not included in the previous reviews and children 0–5 and 5–12 years perhaps are more likely to have similar approaches of learning healthy eating while, due to differences in nutrition requirements and other environmental influences in older age/adolescents, distinct intervention strategies may be needed [33]. *Intervention:* Studies were included where the intervention was inclusive of an experiential learning activity, with one or more of the following characteristics: (1) children played a central role in the activity, allowing them to engage with and explore the phenomena; (2) the activity went beyond the provision of information, purely instruction, encouragement, equipment or change to the environment; (3) the activity required children’s input and children had to physically do something as part of the learning activity; (4) the children had a level of autonomy in completing some part of the activity that required them to be creative, problem-solve, be reflective; (5) the activity invoked the children’s thoughts as well as a sense of taste, touch, smell, feel; (6) the activities were specifically designed to have a learning experience that enhanced healthy eating measured post-intervention; (7) the activities had a clear learning task or skill as the outcome; (8) the children had direct exposure to the phenomena being studied; (9) the activity was coordinated/ facilitated by a leader such as a teacher or an educator, farmer, or parent; and (10) the facilitator(s) provided the structure for the activity such as basic instruction, posing questions to invoke problem-solving, creative thinking or reflection. The study was excluded if there was no experiential activity for children as part of the intervention. *Setting*: Studies were included from all settings (e.g., school, after school programs, preschools/early childhood education and care centers, farms, and school canteens) and there was no exclusion based on the study setting. *Outcome:* a study was included if it had at least one outcome related to food or nutrition behavior, attitudes, or knowledge. A post hoc protocol deviation was made to exclude studies where the outcome was physical activity because of the higher than anticipated number of studies identified. Studies investigating effects on physical activity outcomes will be reported in a separate review.

### 2.3. Study Selection

Study records were imported into EndNote reference software version X9 (Clarivate Analytics, London, UK). Duplicate studies were removed, and two reviewers (M.L.H. and G.N.) independently screened the titles and abstracts. All studies included by at least one reviewer were then assessed for inclusion by the two reviewers (M.L.H. and G.N.) at the full-text stage. Where discrepancies of inclusion/exclusion existed, discussions were conducted between the reviewers to reach a consensus.

### 2.4. Data Extraction

Data for the included studies were extracted using a standardized data extraction table (Table 2) devised by one reviewer (S.D.V.) and discussed with the author group. The information collected included study authors/year of publication, country of study, study design, theoretical framework used, study sample (size, age of participants), intervention (duration, experiential-based learning activities, outcome measures/tools) and results. A second reviewer (D.P.C.) verified the information extracted to reduce error and bias.

### 2.5. Risk of Bias Appraisal

To assess the potential risk of bias of included studies, the revised Cochrane Risk-of-Bias tool for randomized trials (RoB 2) [34] was independently completed by two reviewers (S.D.V. and Z.Z.), with two additional reviewers (R.A.J. and D.P.C.) consulted if consensus could not be reached. This tool examines five domains: the randomization process; deviations from the intended interventions (effect of assignment to intervention or effect of adhering to intervention); missing outcome data; measurement of the outcome; and selection of the reported results. We used the Revised Cochrane risk-of-bias tool for randomized trials (RoB 2) criteria for overall risk-of-bias judgement [35]. The overall risk-of-bias was judged using the following criteria: (1) low risk of bias—the study was judged to be at low risk of bias for all domains for this result,(2) some concerns—the study is judged to raise some concerns in at least one domain for this result, but not to be at high risk of bias for any domain (3) high risk of bias—the study was judged to be at high risk of bias in at least one domain for this result or the study is judged to have some concerns for multiple domains in a way that substantially lowers confidence in the result [35].

### 2.6. Data Synthesis and Analysis

To enable comparison between studies and estimate the relative magnitude of the effect of the interventions, effect sizes for the difference between the intervention and control groups on each outcome measure (increased intake of fruits and vegetables/decreased consumption of unhealthy foods, increased preference for healthy foods/decreased preference for unhealthy foods, and increased nutritional knowledge) were calculated, regardless of their statistical significance. Firstly, the pooled SD was calculated by using the following equation from Cohen [36]:(1)$$SD{*}_{pooled}=\sqrt{\frac{\left(n_{1}-1\right)SD_{1}^{2}+\left(n_{2}-1\right)SD_{2}^{2}}{n_{1}+n_{2}-2}}$$
where: *SD_1_* is the standard deviation of the intervention group, *SD_2_* is the standard deviation of control group 2, *n_1_* is the size of the intervention group and *n_2_* is the size of the control group. The mean difference between the intervention and control groups was divided by the standard deviation (*SD*) for both groups (pooled standard deviation *SD*). Effect sizes were then calculated using the Cohen’s d formula: d = (M1—M2)/*SD* pooled [37], where M1 is the mean of the intervention group, M2 is the mean of the control group and *SD**_p_* is the pooled standard deviation for both groups.

Finally, the studies were divided into two categories according to the age of the intervention participants; that is, pre-school and primary school and mean effect size was then calculated for each study by dividing the sum of all effect sizes by the number of effect sizes (for healthy/unhealthy foods separately) for behavior, attitude, and knowledge outcomes. Effect sizes were interpreted as small (<0.2), medium (0.2–0.8), and large (>0.8) [36].

## 3. Results

### 3.1. Study Selection

Ninety-eight studies described healthy eating-related outcomes in children. Of these, 52 studies did not have a control group and 21 studies were non-randomized controlled trials, thus were excluded. In total, 25 eligible studies were included in the final review, as shown in Figure 1.

### 3.2. Study and Intervention Characteristics

The characteristics of the studies and outcomes are summarized in Table 2. Of the 25 included studies, nine were RCTs and 17 were CRCTs. Six studies involved children aged 0–five years and 19 studies involved children aged six–12 years. Most of the included studies (16/25) were conducted in the United States, with the remainder in England (*n* = 1), Spain (*n* = 2), Norway (*n* = 2), Belgium (*n* = 1), Lebanon (*n* = 1), Nepal (*n* = 1) and Bhutan (*n* = 1). A number of studies were underpinned by a number of different theoretical frameworks including Social Cognitive Theory (*n* = 8, [45,47,48,49,50,53,54,61]), Social Learning Theory (*n* = 1, [44]), Intervention Mapping Protocol (*n* = 1, [38]) Bronfenbrenner’s Ecological Theory (*n* = 1, [43]), Chronic Care Model (*n* = 1, [41]) and Social-Ecological Model (*n* = 1, [51]). Though nearly half of the studies (12/25) did not report the use of any theoretical model in the intervention development [39,40,42,46,52,55,56,57,58,59,60,62].

The majority (18/25) of studies were conducted in the primary school setting [44,45,46,47,48,50,51,52,53,54,55,56,57,58,59,60,61,62], six in pre-school settings [38,39,40,41,42,43] and one in a non-school or education setting, namely scout camps [49]. Nearly three-quarters of the studies (16/25) had a high risk of bias. Eight studies were graded as having ‘some concerns’ [39,49,50,52,56,57,60,62] while only one study was rated as having a low risk of bias [42]. Overall, the included intervention studies had low methodological quality due to three of the domains consistently being rated low quality for most of the included studies, which may impact the validity of the results. The assessment domains that consistently were rated as low quality included missing outcome data, risk of bias in the measurement of the outcome and risk of bias in the selection of the reported result (see Appendix A). Of the 25 studies, nine [39,40,43,45,47,48,51,58,61] studies involved parents directly in the intervention activities with children. Of the nine studies that directly involved parents, three were conducted in the preschool setting.

### 3.3. Experiential Learning Activities

Nine types of experiential learning activities were used across the 25 studies, which included: (1) Taste-testing (i.e., children tasting food products (*n* = 19)); (2) Games (i.e., guessing food, food labelling competitions, card/board games, fun play, mystery bag (*n* = 8)); (3) Creative/art activities (i.e., coloring, drawing, collage, portraits, art and craft on fruits and vegetables, fruit and vegetable charts, posters/pamphlets (*n* = 10)); (4) Storybooks (i.e., making food-related stories (characters) (*n* = 6)); (5) Shopping list development and food purchasing (i.e., creating a shopping list, selecting food/meals, simulated shopping and food classifications, imaginary trips to supermarket and gardens (*n* = 7)); (6) Food preparation and cooking/preparing foods, fruit and vegetables, snacks and other foods/meals (*n* = 7)); (7) Calculations/recording (i.e., sugar and fat, veggie math, three-day fruit and vegetable intake, the personal food pyramid and other math activities with food (*n* = 5)); (8) Sensory evaluation (i.e., smell, feel, sight and sound of foods (*n* = 4)); and (9) Gardening (i.e., planting and harvesting of fruits or vegetables (*n* = 2)) (see Appendix A).

The types of activities used in interventions with preschool-aged children (in early childhood education and care settings) and with primary school-aged children (in primary schools and community settings) were mostly similar, although activities used in early years education and care settings were targeted to earlier developmental stages using sensory play, storybooks, songs, and creative art activities. Of the six studies with preschool children, four studies focused entirely on experiential learning activities [38,40,42,43] while two studies [39,41] combined experiential activities with nutrition education lessons (i.e., a theory-based component). Of the 19 studies conducted with primary school-aged children, seven studies focused entirely on experiential learning activities [45,46,47,49,56,57,62] while 12 studies combined experiential activities with nutrition education [44,48,50,51,52,53,54,55,58,59,60,61].

### 3.4. Intervention Effects

The effect sizes of the intervention (experiential learning activities) on the outcomes; behavior, attitudes, and knowledge (healthy foods and unhealthy foods) are presented in Table 3.

Table 3 highlights the experiential learning activities and effect sizes on outcomes which were grouped as healthy foods and unhealthy foods. (Healthy foods/ Unhealthy foods = see Table 2).

In preschool-aged children, five studies [38,40,42,43] measured behavior change. Two of these [42,43] reported small effects for healthy foods (increasing fruit and vegetable consumption), with both involving sensory evaluations such as feeling/touching fruits and vegetables, or food drawing and coloring activities and games and [43] involved parents in the intervention activities. Only one study in this age group measured changes to children’s attitudes towards healthy foods [42]. This study reported a statistically significant but small mean effect (M*_d_* = 0.23) on changing preschool children’s preferences for, and self-efficacy and willingness to taste, unfamiliar fruits and vegetables. This intervention used multiple experiential learning activities including taste-testing, sensory evaluation, games, storybooks, and creative/art activities. One study measured change in nutrition-related knowledge [39] but effect sizes could not be calculated due to missing data.

In primary school-aged children, sixteen studies [44,45,46,47,49,50,51,53,54,56,57,58,59,60,61,62] measured food-related behavior change. Effect sizes were able to be calculated for eight studies in which the duration of intervention lasted between two and eighteen months [47,48,51,52,53,54,57,58]. Two of these [47,61] had large, significant mean effects (M*_d_* = 1.0) in relation to healthy foods (fruits and vegetable intake). Multiple combinations of experiential learning activities were reported by these studies, including games, role-playing food preparation/cooking, school gardens, and taste-testing. Two of the studies [47,61] with high effects on increasing intake of healthy foods had involved parents directly in the intervention activities. One study [61] also combined experiential learning activities with nutrition education classroom lessons. Three additional studies [56,59,62] reported medium effects (M*_d_* = 0.4). One of these studies [59] was moderately effective in increasing the consumption of healthy foods (fruits and vegetables) through school gardening and taste-testing over one school year however they also included nutrition education lessons. The other two studies [56,62] were moderately effective in reducing consumption of unhealthy foods and used a range of experiential learning activities such as food preparation/cooking, taste-testing, games, creative art activities, sensory evaluation [62] and one study additionally used simulated food purchasing [56].

The remaining seven studies [48,49,51,53,54,57,58] had small (M*_d_* = 0.2) but significant effects for increasing consumption of healthy foods (fruits and vegetables) [48,57,58] or reducing consumption of unhealthy foods such as sugar-sweetened beverages [49] chips and sugar-sweetened drinks [54], sweet snacks, fast foods [53] or intakes of sodium, sugar, and total calories [51]. Six [48,49,51,53,54,57] of the seven studies used a range of experiential learning activities, such as food preparation/cooking, taste-testing, games, songs, creative/art activities, storybooks and, role-playing, while one [58] focused only on gardening. Most of these studies (5/8) [48,49,51,53,54] combined experiential learning with nutrition education lessons. Half of these studies (4/8) [48,49,54,57] had an intervention duration between three to six months, while three studies [51,53,58] had a duration ranging between one to two years.

Eight studies [47,49,52,54,56,57,58,62] conducted with primary school-aged children measured changes in children’s attitudes towards healthy eating, such as self-efficacy and willingness to try new foods. One of these studies [58], involving a two-year school-based gardening program, had a large significant effect (*d* = 1.12) on increasing attitudes related to healthy eating (preferences and self-efficacy for choosing fruits and vegetables). Three studies [47,49,54] had a medium significant effect (M*_d_* = 0.7) on improving attitudes towards healthy and unhealthy foods. All studies that reported large or medium effects used a range of experiential learning activities including food preparation, taste-testing, games, role-plays, and storybooks. The duration of these interventions was between two to four months. One study [49] was conducted in a scout camp setting, another [47] included home-based activities, and one [54] included nutrition education lessons in a school classroom. The remaining four studies [52,56,57,62] had small effects (M*_d_* = 0.29) on changing preferences and self-efficacy for healthy foods, including choosing/liking of fruits [57], fruits and vegetables [52,62], and willingness to choose unfamiliar fruits and vegetables [62]. All four studies used a variety of experiential learning activities, such as simulated food purchasing, food preparation, taste-testing, games, storybooks, and creative/art activities. These studies were of short duration, ranging from single sessions to a period of four months. Two studies [54,58] combined nutrition education classroom lessons with experiential learning activities.

Ten studies [39,47,48,51,52,53,54,55,57,58] measured change in primary school-aged children’s knowledge regarding food, nutrition or healthy eating. Of the eight studies for which effect sizes could be calculated [47,48,51,52,53,54,57,58], four [52,53,54,58] had a large effect (M*_d_* = 1.1) and reported significant effects on increasing knowledge about healthy eating (nutrition, fruits, and vegetables). Of these four studies, three [52,53,54] used a range of experiential learning activities, such as food preparation, taste-testing and games combined with nutrition education lessons. One of the studies [58] included only gardening. Two of these studies were conducted over one to two months [52,53,54], while two took place over one to two years [47,59]. The remaining four studies [47,48,51,57] had a small effect (M*_d_* = 0.01) on increasing children’s knowledge of healthy foods and these studies used a range of experiential learning activities, such as food preparation/cooking, taste-testing, games, songs, creative/art activities, storybooks, and role-playing. Three studies [48,51,57] combined nutrition education with experiential learning activities. The duration of the interventions ranged between two to six months [47,48,57] and two years [51].

## 4. Discussion

### 4.1. Main Findings

The purpose of this systematic review was to examine the effectiveness of experiential learning interventions conducted in pre-schools, primary schools, and community settings for improving healthy eating related knowledge, attitudes, and behaviour in children aged birth to 12 years. Interventions with pre-school aged children that applied strategies such as sensory evaluation activities, playing games, storybooks, role-modelling and creative art activities tended to have a large effect on food behaviours and attitudes. However, there were fewer studies conducted in preschool-aged children compared to older children and the effects were smaller, therefore less evidence of effective experiential learning approaches was found for this age group. Most of the included intervention studies were conducted in the primary school setting, and those that used strategies such as food preparation/cooking, taste-testing, games, role-playing, and gardening, had the greatest effect across the three outcomes (behaviour, attitude, and knowledge) in this age group. There was only one study conducted in a community setting (i.e., a scout’s camp) and it reported a small intervention effect.

The majority of the included studies had used a combination of experiential learning approaches; therefore, the impact of individual experiential learning approaches could not be established. However, a few approaches showed promise and were typically used across the most effective studies. For instance, gardening showed a large effect across the three outcomes (increasing knowledge, preferences and consumption of fruits and vegetables) in studies among primary school-aged children [53,58]. The exception was one study [59] which reported a very low effect due to the reported small study sample. Our findings on the effectiveness of gardening strategy are consistent with other studies among primary school children [63,64], however, these studies had compared an active comparison group with gardening (teacher-led versus expert-led gardening) instead of a control group.

Taste-testing was also commonly used in studies across both age groups and demonstrated a large effect on behavioral outcomes [42,61] however, it was often applied together with sensory evaluation, food preparation, cooking and/or gardening. The exceptions were two studies that had included taste-testing in their intervention and reported a small effect [38,43]. However, these two studies had (1) a lower intervention intensity (six weekly group educational sessions), (2) a low adherence by the intervention group, (3) combined nutrition education with the experiential learning activities, and (4) a smaller sample size (e.g., <100). Previous studies that investigated the effectiveness of taste-testing in primary school curriculum have recommended using experiential learning approaches for desired outcomes [65]. A recent scoping review that examined children’s involvement in meal preparation and the associated nutrition outcomes also found that hands-on meal preparation can instil positive perceptions towards nutrition/healthy foods, and potentially improve children’s diet [66]. Hence our finding is consistent with the existing literature.

Creative art activities such as coloring, drawing, making a collage using food pictures, portraits, art and craftwork and charts on fruit and vegetables, making posters and pamphlets were also utilized consistently in studies with large effect sizes across the age groups and outcomes. These activities were effective when used in combination with other strategies such as cooking and taste-testing. Art and craft activities linking colors (rainbow) with fruit and vegetables possibly broadened children’s knowledge and awareness of eating a variety of fruits and vegetables [67]. However, there is a lack of existing supporting evidence on this potential influence.

In relation to children’s dietary behavior changes, studies that focused on both healthy and unhealthy foods were effective. However, for changing attitudes and knowledge, interventions that focused on providing positive messages related to increased consumption of healthy foods tended to be more effective than those that focused on discouraging unhealthy foods. Furthermore, studies in primary school children with medium to large effects reported using used Social Cognitive Theory SCT in their intervention development [47,53,61]. However, these studies did not specify how the concepts of SCT concepts were incorporated. These interventions may have been effective because SCT explains how children can acquire and maintain behaviour patterns and that behavior, personal and environmental factors interact to describe and predict behaviour change in a reciprocal way [68]. Self-efficacy, outcome expectation, skill mastery and self-regulation are the key concepts of social cognitive theory that can be used to explain and predict behaviour changes [69] Furthermore, knowledge gained through direct involvement in experience is integral to experiential learning [27]. This central idea is found in a range of theories and outlines surrounding experiential learning.

The studies with a short-term intervention duration (up to twelve weeks) for preschool-aged children were also more effective compared to those of longer duration (up to six months) for demonstrated behavior change. However, these studies did not report any follow-up assessments thus it is unclear whether the effects were only short-term or if longer-lasting benefits were produced. The exception was one study [47] that conducted a follow-up assessment eight months after the intervention and reported that effects were maintained.

Regarding the interventions with preschool-aged children, the strategies that seemed more suitable to their developmental stages were effective. The two most promising programs included a study by Dazeley et al. [42] which used sensory evaluation activities (use of senses), especially with fruits and vegetables and reported a large effect. The other study was by Jisoo et al. [43], which focused on storybooks (with visuals) and involved parents completing activities with their children. Younger children perhaps acquire their food preferences by direct contact with foods through sensory experiences such as tasting, feeling, seeing, and smelling foods [70] which might explain why this strategy is effective for this age group. Our finding is similar to recent research [71] which also showed positive results from the exposure to pictures of foods in toddlers.

In contrast, it was also evident that some of the intervention studies that reported smaller effects also used similar experiential learning approaches to those of more effective studies. However, a range of other possible factors, beyond the intervention strategies, may have influenced their relative impact. For example, these studies tended to use child reports of eating behaviors, did not use validated tools to measure behavior, had extended durations but with lower intensity of intervention strategies (e.g., infrequent intervention sessions), and combined experiential learning activities with more didactic classroom sessions.

The relative effectiveness of school-based experiential learning approaches to promote healthy eating in children compared to nutrition education alone was supported in an earlier systematic review and meta-analysis by Dudley et al. [22] They examined the teaching strategies of 49 interventions that reported on healthy eating outcomes for primary school children and found that experiential learning strategies had the largest effects across all outcomes. However, that review did not focus on the effectiveness of different types of experiential learning activities and only included studies conducted in the primary school setting. Similarly, another review by Charlton et al. [30] that focused on school-based experiential learning and nutrition education interventions among primary school children found that interventions that included multiple or a combination of experiential learning strategies increased children’s preferences for, knowledge of, and consumption of healthier foods. Both the earlier reviews included quasi-experimental study designs as well. Our findings extend these reviews by examining only RCT interventions and including pre-school and community settings.

### 4.2. Implications for Interventions

This review suggests that the experiential learning interventions may be more successful to the extent that they (a) include multiple or a combination of experiential learning strategies in the intervention, indicating that the more diverse the intervention, the more likely it was to be successful, (b) involve parents in the intervention activities, such as models in cooking and gardening which may create awareness, reinforce knowledge gained and encourage healthy behaviours [72], (c) make strategies fun, interesting, realistic, and more engaging for children, which demonstrates the importance of experiential learning (hands-on activities) as they involve processes where the learners actively experience activities, attempt to conceptualize what is observed, and reflect on those experiences [27], (d) are grounded on an effective behaviour change theory such social cognitive theory, (e) are focused on specific and targeted food behaviours for improvement such as “choose vegetables as snacks”.

### 4.3. Implications for Future Research

Based on this review there are recommendations for future research in this area. There are fewer experiential learning healthy eating interventions conducted in pre-school and community settings compared to primary school, thus more studies are needed in these settings. Most of the studies were overall rated as having low methodological quality due to the following factors being consistently rated as low quality: missing outcome data, risk of bias in the measurement of the outcome and risk of bias in the selection of the reported results. Future researchers could focus on addressing these limitations to enhance the quality of the evidence. Our review supports the need for more among preschool-aged children and for settings beyond primary schools, such as communities. Furthermore, most of the included studies did not report the use of any theoretical model in the intervention development, thus it is recommended that future interventions are built on a behavioural theory. For effectiveness, future studies should consider conducting follow-up assessments to understand if intervention effects are maintained. Development of short and intense interventions that are better suited for the specific settings.

### 4.4. Implications for Policy

For primary school-based experiential learning interventions, to deliver our recommendations to policymakers, factors such as cost, context, dose-response, and sustainability of the intervention should be considered. Policymakers should also focus on specific school food environment policies that improve targeted dietary behaviours such as healthy eating [73].

### 4.5. Strengths and Limitations

The current review updates and extends the previous reviews and includes studies with a broader age group and interventions delivered outside of school settings. We used broad search terms and a comprehensive inclusion criterion, which yielded many eligible studies which were independently screened by two reviewers. Only RCT and CRCT studies with experiential learning interventions were included, which enhances the internal validity of the review. We calculated effect sizes (Cohen’s d) to quantify the relative effect of the intervention strategies on the outcomes as well as the relative effect on healthy and unhealthy choices across age groups, which has not been done previously. We also assessed the risk of bias using the Cochrane Collaboration tool, which was important for highlighting methodological gaps in the evidence base.

There were a few limitations associated with this review. This review only included papers published in English, therefore we may have not included papers published in other languages. Our review only included RCTs and CRCTs, but not quasi-experimental studies, which would have strengthened the internal validity of our review. However, in relation to studies conducted in schools, it may not always be possible to randomize groups to intervention or control conditions (e.g., if schools were building gardens). Evidence from such studies with less robust designs may still provide useful information about the effectiveness of experiential learning interventions. We were not able to calculate effect sizes for some studies, despite best efforts to obtain further information from study investigators. Given that only one study was conducted outside of the school setting, there was a limited ability to identify effective experiential learning activities for other settings. Likewise, many of the interventions were conducted with school-aged children, rather than with younger age groups. The risk of bias assessments of the studies was generally high, therefore the strength of the conclusions from this review may need to be considered carefully. Lastly, this review is limited to the effects of experiential learning activities on healthy eating outcomes only, and therefore findings are not generalized to other lifestyle behaviours such as physical activity.

## 5. Conclusions

Experiential learning activities are a useful strategy to improve children’s knowledge, attitudes, and behaviors towards healthy eating. Strategies such as food preparation/cooking, taste testing, playing games, role-playing, and gardening were found to positively affect nutrition outcomes for primary school-aged children. For preschool-aged children, strategies such as sensory evaluation, taste-testing, interactive games, creative arts activities, and storybooks hold promise, but more research in this age group is needed. Key features of successful interventions included combining multiple strategies, involving parents, being grounded on a theoretical model and delivering shorter but more intense interventions. The findings of this review provide useful insight for future interventions that seek to apply experiential learning to the improvement of healthy eating in children.

## Figures and Tables

**Figure 1 ijerph-18-10824-f001:**
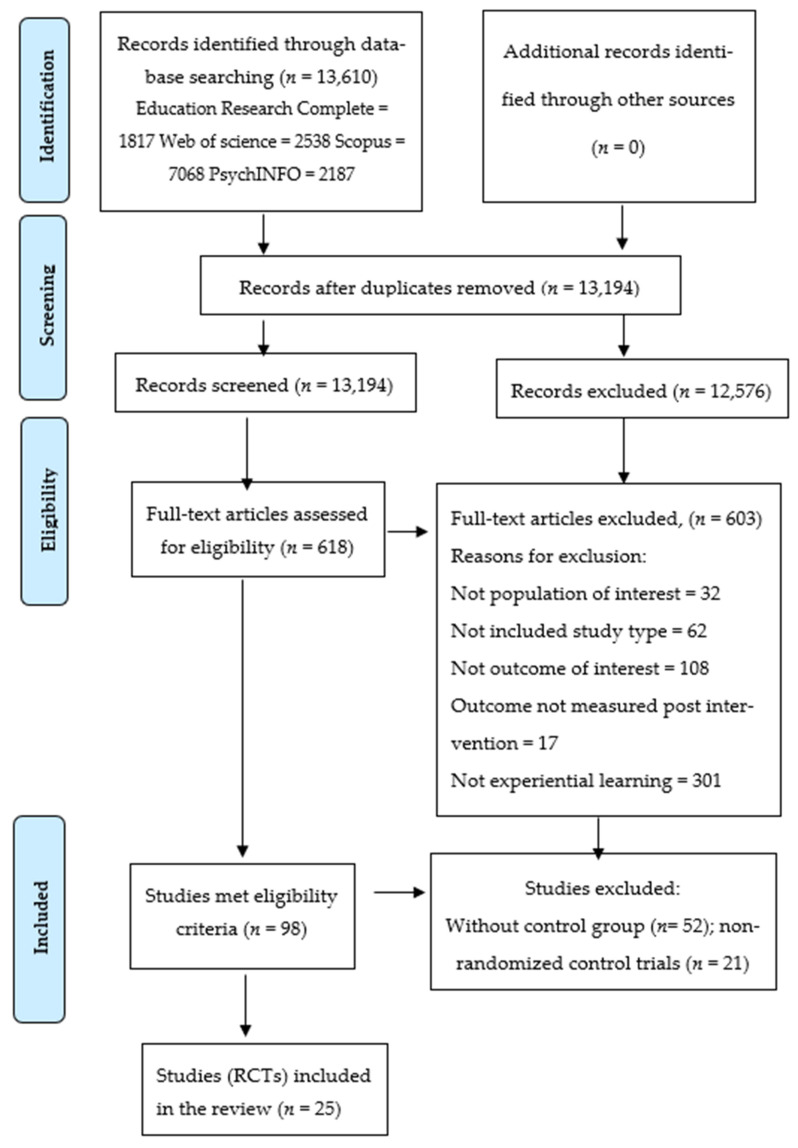
Flow chart of the study selection process. (PRISMA flow diagram [32]).

**Table 1 ijerph-18-10824-t001:** Search terms.

PICO	Booleans	Search Terms
Population		“Child*” OR “Preschool” OR “Elementary school” OR “Elementary student” OR “Elementary education” OR “Grade 1” OR “Grade 2” OR “Grade 3” OR “Grade 4” OR “Grade 5” OR “Grade 6” OR “Kindergarten” OR “Primary education” OR “Primary school” OR “Early years”
Intervention	AND	“Play-based learning” OR “Learning through play” OR “Experiential learning” OR “Learning centered play” OR “Student-centered learning” OR “Guided play” OR “Facilitated play” OR “Play-based education” OR “Play education” OR “Educati* Activ*” OR “Interactive learning” OR “Playful pedagogy” OR “Active learning” OR “Experiential education” OR “Experience-based learning” OR “Program*” OR “Intervention” OR “Workshop” OR “Promotion” OR “Project”
Outcome	AND	“Nutrition*” OR “Food” OR “Diet*” OR “Eating habits” OR “Fruit” OR “Vegetable” OR “Healthy eating”

**Table 2 ijerph-18-10824-t002:** Experiential learning interventions and healthy eating outcomes in children.

Authors (Year) Country	Study Design/ Theory	Sample Size, Age/Grade Involved Parent	Setting, Duration, Experiential Learning Activity	Measures/ Tools	Results	Overall Risk of Bias
**Studies conducted in a preschool setting (<5 years old children)**						
Vereecken et al., (2009) Belgium [38]	RCT IMP	*N* = 1063 2–3 years No	Preschools. Six months. Tasting food, role- modelling (story and characters)	Changes in consumption of fruit, vegetables, snacks (pastry, savoury snacks, and sweets) and drinks. Observations recorded by teachers and parent-reported using FFQ.	I > C for children’s fruit consumption (parental reported), due to an increase in fruit made available at school 0.11 (95 % CI: 0.00, 0.21) *p* < 0.044) and not due to an increase in fruit brought from home (intervention effect = –0.02 (95 % CI: −0.13, 0.08) *p* = 0.677). I = C for other food items (snacks, vegetables, and beverages). **(↑ +)**	High
Witt et al., (2012) U.S. [39]	CRCT NR	*N* = 268 4–5 years Yes	Preschool-based. Six weeks: 2 x 15-30min lessons + 1 imaginary trip per week. Fun, interactive activities, songs/music, colour, appeal to senses, role plays, healthful eating, food tasting	Weighed snack consumption of fruit and vegetables during childcare. 3-day food diary, FFQ.	I > C for all. Post-test: Fruit - d = 1.29, *p* < 0.001; vegetables - d = 0.90, *p* < 0.001 follow-up: Fruit - d = 0.68, *p* < 0.001; vegetables - 1.20, *p* < 0.001 **(↑ +)**	Some Concerns
Brouwer and Neelon, (2013) U.S. [40]	RCT NR	*N* = NR. (Average 19 children × 4 centres) 3–5 years. Yes	Child-care centres. Four-month gardening program to yield one crop per month and tasting produce.	Increase in no. of V & F provided to & consumed by children in childcare. Observation (meals and snacks). Recording (all foods/beverages served, consumed, and wasted). Nutritional value and food groupings (USDA MyPlate).	Post-intervention, Vegetable consumption, (mean intake) I (0.25 (1.10)) > C (−0.18 (0.52)). Fruit consumption, (mean intake) I (−0.33 (0.72)) < C (0.15 (0.25)). **(↑ +)**	High
Martínez-Andrade et al., (2014) U.S. [41]	CRCT CCM	*N* = 306 2–5 years No	Primary care clinics. Six weeks x 2 h. 90 min - nutrition education, 30 min- preparation and consumption of healthy foods, calculating the quantity of sugar/ fat in processed foods, creating shopping lists.	Parent-reported (three- and six-month follow-up). Dietary intake- Child FFQ	I > C for vegetable consumption: 6.3 servings/week, (95% CI, 1.8, 10.8) at 3 months. I = C for behaviour at 3 and 6 months. At 3-month sweet snacks (−3.9 servings/ week; 95% CI, −8.9, 1.1), sugar added to drinks (−2.2 Servings/week; 95% CI, −8.4, 4.1), and effects reduced at 6 months. **(↑+ ↓-)**	High
Dazeley and Houston-Price, (2015) England [42]	CRCT NR	*N* = 92 1–3 years No	Day-care nurseries. Once/ day x four weeks. Four activity sheets in total, each with three games specific to senses: sight, smell, touch or sound and activities such as drawing, colouring, storybooks nursery rhyme and taste testing.	Researchers recorded foods touched and tasted by children (video camera) and online coding.	I > C, children touched and tasted more of the vegetables to which they had been familiarised in their playtime activities than of a matched set of non-exposed foods t (53) = 2.05, *p* = 0.046). **(↑ +)**	Low
Jisoo et al., (2018) U.S. [43]	CRCT BET	*N* = 42 4–5 years Yes	Preschool & home. Family backpack (hands-on activities/supplies) distributed over 12 weeks. Children’s picture book, hands-on activities (1) “Mystery Bag,” math activity on F&V by touch; (2) “My Favourites’,” art activity on F&V by drawing (3) “Graphing F&V”	Parent-reported children’s fruit and vegetable consumption FFQ	I > C for children’s fruit and vegetable consumption. [t (21) = 2.49, *p* < 0.05 for fruits; t (21) = 3.92, *p* < 0.01 for vegetables]. **(↑ +)**	High
**Studies conducted in the primary school and community setting (6 to 12 years old children)**						
Perry et al., (1998) U.S [44]	CRCT SLT	*N* = 441 Grades 4–5 No	Primary schools. Two curricula: “High 5” and “5 for 5,” Each included, 16 × 40–45-min classroom sessions, 2 × a week for eight weeks. Skill-building, problem-solving activities, snack preparation & taste testing. Curricula introduced; role models (comic books in High 5), adventure stories (5 for 5), competitions (eating fruits & vegetables).	Lunchroom observations and 24-h food recall measured food consumption. Parent telephone surveys and a health behaviour questionnaire (psychosocial factors).	Lunch observations: I > C for vegetable consumption in girls (Δ = 0, 26 servings, *p* < 0.05) than boys (Δ = 0, 04). 24-Hr recalls. I > C for servings of fruits & vegetables per 1000 kcal, and servings of fruit per 1000 kcal. Health Behaviour: Teacher perceived- I > C for eating, need to eat, reports of asking for, daily servings of fruits and vegetables. **(↑ +)**	High
Bere et al., (2005) Norway [45]	CRCT NR	*N* = 517 12–13 years Grade 6 Yes	School-based. Two school years; baseline to follow-up 1 = 8 months and follow-up 2= 20 months. Preparing fruit and vegetables (snacks), taste testing.	Self-reported fruit and vegetable intake (24-h fruit and vegetable recall).	I > C for fruit and vegetable intake at follow-up 1 and follow-up 2 (effect sizes = 0.6 and 0.5), *p* = 0.07) at school and all day. **(↑ +)**	High
Bere et al., (2006) Norway [46]	CRCT SCT	*N* = 369, 12–13 years Grade 6. No	School-based. One school year. Preparing fruit and vegetables (snacks), taste testing, information session, monitoring own fruit and vegetable intake for three days, self-assessment and goal setting for future intake.	Self-reported fruit and vegetable intake (24-h fruit and vegetable recall).	I = C for intake of fruit and vegetables eaten at school or all day, neither at follow-up 1 (22% versus 15% subscribers) nor follow-up 2 (15% versus 26% subscribers), *p* = (0.51, 0.76 0.41). **(↑ +)**	High
Chen et al., (2009) U.S [47]	RCT SCT	*N* = 67 8–10 years. Yes	Family-based. Once/ week for eight weeks, interactive fun activities (games & play), role-playing selecting healthy meals, food choices-high sugar, high-fat foods, interactive food preparation. Family component (two 2-h sessions)	Self-reported. Dietary intake of children (3-day food diary) and food choices (Health Behaviour Questionnaire)	I > C for the decrease in fat intake and increase vegetable and fruit intake. (*p* < 0.05) **(↑ +) (↓−)**	High
Fulkerson et al., (2010) U.S. [48]	RCT SCT	*N* = 44 8–10 years. Yes	Elementary School. Five × 90-min sessions, six months - interactive nutrition education, taste testing, cooking skill building, parent discussion groups, and hands-on meal preparation.	Child food preparation skill (Questionnaire). Home food availability/meal offering (Inventories). Dietary assessment (24-h diet recalls).	I > C for food preparation skill development (*p* < 0.001), consumption of fruits and vegetables (*p* < 0.08), and intakes of key nutrients (all *p* values < 0.05). **(↑+)**	High
Rosenkranz et al., (2010) U.S [49]	CRCT	*N* = 76, 9–12 years. No	Girl Scouts and home. Educational curriculum, FV snack preparation, role-playing, tasting of FV snacks, promotion of FV consumption and prohibition of SSB, candy over 4 months.	Troop leader health promotion behaviours and environmental opportunities for healthful eating in the troop meetings.	I > C for all. Opportunities for healthful eating (d= 210.8, *p* < 0.001), promotion of healthful eating (d = 18.14, *p* < 0.001). 1< C for discouraged healthful eating (*p* = 0.002) **(↑+ ↓−)**	Some Concerns
Keihner et al., (2011) U.S [50]	RCT SCT	*N* = 1154 8–12years Grades 4–5. No	Elementary schools. 10 × 50 min classroom-based nutrition education lessons (grade-specific) with activities including Fruit and Vegetable rap songs, serving size poster, and stickers over the eight weeks.	Pre/post surveys measured knowledge, outcome expectations, and self-efficacy (SE) using a questionnaire.	I > C for Fruit and Vegetable knowledge (4 items, *p* < 0.05 to *p* < 0.001); positive outcome expectations (fifth grade only, *p* <0.001); asking/shopping and eating Self Efficacy (*p* = 0.04 and *p* < 0.001). **(↑ +)**	Some Concerns
Katz et al., (2011) U.S [51]	CRCT SEM	*N* = 1180 7–9 years. Grades 2–4. Yes	Elementary School. Four × 20-min sessions over two school years. (Minilessons) on food choice and health, interactive activity/ hands-on ‘‘spying on food labels’’ game, emphasising healthy choice and summarising key points.	Nutrition Knowledge (food label literacy/ food choices) – a standardised test instrument. Dietary Pattern- Youth and Adolescent Questionnaire and (FFQ).	I > C for nutrition knowledge (*p* < 0.01). Grade 3 students showed the most improvement, 23% (*p* < 0.01). I = C for improvements in dietary patterns, in terms of total caloric, sodium, and total sugar intake (*p* > 0.05). **(↑+ ↓−)**	High
Wall et al., (2012) U.S [52]	CRCT NR	*N* = 1937 9 years Grade 4. No	Elementary school. Four lessons × 3–5 weeks of learner-centred activities; vegetable tastings, worksheets, handouts.	Food preference, attitude, and self-efficacy survey items (from SNAP-Ed intervention)	I > C for vegetable-related preference, attitude, self-efficacy, and knowledge (*p* < 0.001). (Intervention 1.56 ± 5.80); (control 0.08 ± 4.82). **(↑ +)**	Some Concerns
Brown et al., (2013) U.S [53]	CRCT SCT	*N* = 1619 11–14 years. Grades 4–8. No	School-based. Four lessons on calcium and osteoporosis. Prevention and taste-testing food items within two weeks over one academic year.	Interest in lessons, enjoyment of food tasting, eating attitude, tasting experience, new knowledge (21-item survey)	For all foods tasted, students who did not enjoy the food tasting were less interested in the lesson than students who did enjoy the food tasting (all *p* < 0.001). **(↓−)**	High
Habib-Mourad et al., (2014) Lebanon [54]	RCT SCT	*N* = 387 9–11 years. Grades 4–5. No	Primary school. 45 min classroom sessions per week for 12 weeks (10–15 min discussion on nutrition, 30 min - interactive activities (games, hands-on activities- posters, pamphlets, activity booklets, card & board games), food preparation.	Dietary habits, nutrition knowledge and self-efficacy (Questionnaire)	I > C for purchasing and consuming less chips and sugar sweetened beverage (SSB) (86% & 88%, *p* < 0.001) and knowledge and self-efficacy (+ 2.8 & +1.7, *p* < 0.001). **(↓−)**	High
Wells et al., (2015) U.S. [55]	RCT NR	*N* = 3061, 6–12 years, Grades 2, 4, & 5. No	Elementary school. 40 lessons x 60 min (20 for grade two to three, 20 for grade four to six/ week for two years (classroom & garden) Garden activities- planting, harvesting, and tasting as snacks.	Effect school gardens on children’s science knowledge (Nutritional Science Questionnaire)	I > C for science knowledge from wave 1 to waves 2, 3, 4 (*p* < 0.0001), and for dose response (*p* < 0.0001). **(↑+)**	High
Allirot et al., (2016) Spain [56]	RCT NR	*N* = 137 7–11 years. No	Primary school. Single session × 2 h. 1hr cooking workshop- preparing three food items/ chance to see, smell and touch taste ingredients. 1hr creative workshop- collage session (food images-fruits/ vegetables) creating portrait (cutting & gluing food images, making stories with (created characters), drawings (whiteboard), playing games (guessing & drawing). Food selection (familiar/unfamiliar), tasting.	Willingness to choose and taste unfamiliar foods/food intake estimation (Photographs). Liking of the food items (electronic 5-point facial scale) Subjective hunger and satiety (Bennet and Blisset’s “Teddy the Bear” hunger and satiety scale).	I > C for mean number of unfamiliar foods chosen per child (*p* = 0.037), for willingness to taste the unfamiliar foods (*p* = 0.011), liking for the whole afternoon snack (*p* = 0.034), for 2 of 3 unfamiliar foods and for 1 of 3 familiar foods (*p* < 0.05). I = C for overall food intake and hunger/satiety scores. **(↑+ ↓−)**	Some Concerns
LaChausse, (2017) U.S. [57]	CRCT NR	*N* = 275 Grades 4–6. No	Primary schools. Onex 30–40min session per month × four months.14 HOTM activities included fruit and vegetable tastings, student workbooks, storybooks related to a monthly fruit or vegetable, farm-to-school presentations, and cafeteria posters.	Self-reported. Eating behaviours (Youth Network Survey). Fruit (F) and vegetable (V) preferences (F/V Preferences Scale). Knowledge on F &V- (5 items from General Knowledge Survey). Self-Efficacy to i) Ask for F and V ii) to Prepare F and V iii) to Eat fruits and vegetables.	I = C for both: fruit consumption (b = 0.14, t = 0.89, *p* = 0.38), vegetable consumption (b = –0.17, t = –0.73, *p* = 0.47). I > C for fruit & vegetable preferences, (b = 3.41, t = 2.19, *p* = 0.04) I = C for knowledge of fruits and vegetables, (b = 0.13, t = 0.77, *p* = 0.45). Self-efficacy, I = C for all (to ask for, prepare and eat fruits and vegetables. **(↑ +)**	Some Concerns
Schreinemachers et al., (2017) Nepal [58]	CRCT NR	*N* = 1275 10–15 years. No	School-based. Two school years. Gardening lessons and hands-on practice (cultivation of nutrient-dense vegetables), lessons on gardening and promotional activities to reinforce lessons and strengthen impact by poster displays.	Awareness of fruit (F) & vegetables (V), knowledge about food & nutrition & sustainable agriculture, preferences & liking for F&V (structured questionnaire with (colour photos and multiple choice). F& V consumption (24-hr recall)	I > C for children’s awareness about F and V, knowledge on sustainable agriculture, food, nutrition and health and their stated preferences for eating fruit and vegetables (*p* < 0.01). I = C for improvements in F and V consumption or nutritional status. **(↑ +)**	High
Schreinemachers et al., (2017) Bhutan [59]	CRCT NR	*N* = 468 9–15 years Yes	School-based. One school year of gardening projects to cultivate nutrient-dense vegetables and weekly lessons on gardening/ nutrition. Promotional activities to reinforce the lessons (poster displays, poem displays on school boards, songs, nutrition charts, vegetable charts, pledges)	Awareness of fruit (F) & vegetables (V), knowledge about food & nutrition & sustainable agriculture, preferences/ liking for F and V (structured questionnaire with (colour photos and multiple choice). F and V consumption (24-h recall)	I > C, for children’s awareness of fruit & vegetables by 18.0 % (*p* < 0.01), their knowledge of sustainable agriculture by 15.2 % (*p* < 0.05), preferences for consuming fruit & vegetables by 9.5 % (*p* < 0.05), children’s probability of consuming vegetables the previous day, 11.7 % (*p* < 0.05) but I = C for number of different fruits or vegetables consumed. **(↑ +)**	High
Keihner et al., (2017) U.S. [60]	CRCT NR	*N* = 3463 8–12 years Grades 4–5. No	Elementary schools. 10 weeks of activities during/after school-weekly fruits &vegetable lessons, biweekly classroom promotions/taste tests; posters displayed in/around schools; weekly nutrition materials for parents.	Child reported fruit and vegetable (FV) intake using a 24-h food diary.	I < C for daily Fruit and Vegetable intake, (Mean difference in change between groups, 0.26 cups, *p* < 0.001) **(↑ +)**	Some Concerns
Scherr et al., (2017) U.S. [61]	CRCT SCT	*N* = 409 9–10 years Grade 4 Yes	Elementary school. Nutrition education, cooking demonstrations, school gardens, family newsletters, health fairs, salad bars, tasting over one school year.	Increase in nutrition knowledge (Nutrition Knowledge Questionnaire). Fruit and vegetable intake (FFQ).	I > C nutrition knowledge (mean d = 2.2; *p* < 0.001), total vegetable identification (mean d = 1.18; *p* < 0.001), vegetable preferences or reported fruit & vegetable intake, self-reported general or diet-related parenting practices. **(↑+ ↓−)**	High
Allirot et al., (2018) Spain [62]	CRCT NR	*N* = 86 8–10 years No	Primary school. Single session × 2 h. 1-h workshop- simulated purchasing of ingredients for the preparation of three unfamiliar foods and classifying them as per recipe: 1-h creative workshop—drawing a dish using vegetable or fruit, oral presentation, personal nutritional pyramid and playing guessing game: consumption of afternoon snack—from six food items.	Willingness to choose and taste unfamiliar foods/Food intake estimation (Photographs). Liking of food items (validated illustrative five-point facial scale). Subjective hunger and satiety (Bennet and Blisset’s “Teddy the Bear” hunger and satiety scale).	I > C for mean number of unfamiliar foods chosen per child (0.70 ± 0.14), (0.19 ± 0.07) (*p* = 0.003) and liking for 1 of 3 unfamiliar foods (*p* < 0.05). I = C for food intake estimation and levels of subjective appetite. **(↑+ ↓−)**	Some Concerns

RCT = Randomized control trial, CRCT = cluster randomized control trial, NR = Not reported, SCT = Social cognitive theory, SLT = Social learning theory: BET = Bronfenbrenner’s ecological theory: SEM = Social-ecological model, CCM= Chronic Care Model, IMP = Intervention Mapping Protocol: FFQ = Food Frequency Questionnaire, HOTM = Harvest of the Month, I = Intervention, C = Control, (↑ +) = Increase in healthy foods - fruits and vegetables, (↓-) = Decrease in unhealthy foods - sweet snacks, sugar-sweetened beverages, chips, and fast foods. FV = Fruits &Vegetables, Δ = change, d = difference, Overall risk of bias = See Appendix A for Risk of Bias ratings on individual criteria.

**Table 3 ijerph-18-10824-t003:** Experiential learning activities and effect sizes on outcomes (Healthy foods and unhealthy foods).

**Preschool-Aged Children**		
**Study**	Experiential Learning Activities	Mean Effect Size (Cohen’s d)
**Outcome: Behaviour**		
Dazeley et al. (2015) [42]	Taste-testing, sensory evaluation, games, storybooks, creative/art activities	Healthy foods 0.13
Jisoo et al. (2018) [43]	Games, storybooks, sensory evaluation creative/art activities	Healthy foods 0.12
Vereecken et al. (2009) [38]	Taste-testing, role modelling	Healthy foods 0.01
		Unhealthy foods 0.03
Martinez et al. (2014) [41]	Taste-testing, food preparation/cooking, calculations, creating shopping lists	Healthy foods −0.12
		Unhealthy foods −0.004
Brouwer et al. (2013) [40]	Gardening, taste-testing	Healthy foods −0.04
Outcome: Attitudes		
Dazeley et al. (2015) [42]	Taste-testing, sensory evaluation, games, storybooks, creative/art activities	Healthy foods 0.23
Outcome: Knowledge		
Witt et al. (2012) [39]	Taste-testing, roleplays, games, songs	Insufficient reported data
Primary school-aged children		
Study	Experiential learning activities	Mean effect size (Cohen’s d)
Outcome: Behaviour		
Chen et al. (2010) [47]	Food preparation, role-playing, games	Healthy foods 1.31
		Unhealthy foods −0.05
Scherr et al. (2017) [61]	Gardening, taste-testing	Healthy foods 0.9
		Unhealthy foods −0.66
Allirot et al. (2016) [56]	Food preparation/cooking, taste-testing, games, sensory evaluation, creative art activities	Healthy foods 0.5
		Unhealthy foods 0.2
Allirot et al. (2018) [62]	Food purchasing, food preparation/cooking, games, taste-testing, creative/art activities	Healthy foods −0.12
		Unhealthy foods 0.4
Schreinemachers et al. (2017) [58]	Gardening	Healthy foods 0.3
Brown et al. (2013) [53]	Taste-testing	Unhealthy foods 0.13
Habib-Mourad et al. (2014) [54]	Food preparation/cooking, games	Healthy foods 0.12
		Unhealthy foods −0.13
Schreinemachers et al. (2017) [59]	Gardening, songs, creative/art activities	Healthy foods 0.09
LaChausse, (2017) [57]	Taste-testing, storybooks	Healthy foods 0.06
Katz et al. (2011) [51]	Games	Healthy foods 0.06
		Unhealthy foods 0.07
Fulkerson et al. (2010) [48]	Taste-testing, food preparation/cooking.	Healthy foods 0
Rosenkranz et al. (2010) [49]	Taste-testing, food preparation/cooking, role-playing	Unhealthy foods 0.04
Perry et al. (1998) [44]	Taste-testing, food preparation/cooking, taste-testing, storybook	Insufficient reported data
Bere et al. (2005) [45]	Food preparation, taste-testing	Insufficient reported data
Bere et al. (2006) [46]	Food preparation/cooking, taste-testing	Insufficient reported data
Keihner et al. (2017) [60]	Taste-testing	Insufficient reported data
Keihner et al. (2011) [50]	Songs	Insufficient reported data
Outcome: Attitudes		
Schreinemachers et al. (2017) [58]	Gardening	Healthy foods 1.12
Habib-Mourad et al. (2014) [54]	Food preparation/cooking, games	Healthy foods 0.8
Rosenkranz et al. (2010) [49]	Food preparation/cooking, role-playing, taste-testing.	Healthy foods 0.7
Chen et al. (2010) [47]	Food preparation/cooking, role-playing, games	Healthy foods 0.5
Allirot et al. (2016) [56]	Food preparation/cooking, taste-testing, sensory evaluation, games, creative art activities	Healthy foods 0.3
		Unhealthy foods 0.3
Wall et al. (2012) [52]	Taste-testing	Healthy foods 0.3
Allirot et al. (2018) [62]	Food purchasing, food preparation/ cooking, taste-testing, games creative/art activities	Healthy foods 0.09
		Unhealthy foods 0.14
LaChausse (2017) [57]	Taste-testing, storybooks	Healthy foods 0.04
Outcome: Knowledge		
Schreinemachers et al. (2017) [58]	Gardening	Healthy foods 1.43
Habib-Mourad et al. (2014) [54]	Food preparation/cooking, games	Healthy foods 1.03
Wall et al. (2012) [52]	Taste-testing	Healthy foods 1.03
Brown et al. (2013) [53]	Taste-testing	Healthy foods 0.9
Fulkerson et al. (2010) [48]	Food preparation/cooking, taste-testing.	Healthy foods 0.2
Chen et al. (2010) [47]	Food preparation/cooking, role-playing, games	Healthy foods 0.2
LaChausse (2017) [57]	Taste-testing, storybooks	Healthy foods 0
Katz et al. (2011) [51]	Games	Healthy foods −0.14
Wells et al. (2015) [55]	Gardening, taste-testing	Insufficient reported data

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
