# Peer review of "Experiential Learning Interventions and Healthy Eating Outcomes in Children: A Systematic Literature Review"

_ijerph, 2021, doi:10.3390/ijerph182010824_

Round 1

Reviewer 1 Report

Dear Authors,

This review is interesting but needs substantial clarification and reworking before considering it for publication.

  • Lines 19, 88: Period of time encompassed by the search: please, specify the starting year chose to be included in this review.
  • Line 41-45: The references used may be relevant, but it is not clear why the authors focused on the Australian context? How about other contexts?
  • The current introduction does not seem to justify the importance of the work or follow a clear logical flow. The lack of understanding of this review is deeply concerning. My main concern is that similar systematic reviews were performed (line 60-71). Why is the work significant? How will this review fill in the current knowledge gap? The context of the review should be explicated. Why experiential learning would be a useful strategy to improve children’s knowledge, attitudes, and behaviors towards healthy eating in these settings, and community setting in particular? The authors seem to not be aware of the theoretical models (i.e. SCT, TPB....etc), which are used most often in policy and environmental interventions to change children's dietary intake. The review is limited to the effects of experiential learning activities on dietary outcomes only. It would be considerably strengthened by the inclusion of other lifestyle behaviour such as physical activity.
  • Line 84, 101-102: Why studies with children with mean age above 12 years were not included? How do you define children/adolescent? What age is defined as infant? Please clarify if this review includes children from birth to age 12 years old (Children 0-12 years).
  • Line 86: "Nutrition-related behaviors" should be clearly defined.
  • Table 1: Population- How about children in community-based settings (i.e., day care centers).
  • I think sections 2.2 and 2.3 could be merged.
  • Table 2: "Overall risk of bias" is unclear to me. Please clarify how you rate the strength of study outcomes in a separate table. For example, studies with a ‘++’ rating in at least two of the aforementioned critical domains were defined as low RoB.
  • Table 2: should be "Studies conducted in preschool and community settings (< 5 years old children).
  • Figure 1: It would be good if authors would include the number of record through selected databases (i.e., Scopus=200, web of science=500…etc).
  • Line 197 and Table 2: ''some concerns''- meaning unclear, please clarify.
  • The supporting literature in discussion could benefit from a more recent scan of the literature.
  • I miss a discussion on the effectiveness of experiential learning interventions on improving healthy eating in community-based setting. There is also no discussion of behavior-change theories and models with a focus on SCT, SLT, BET, SEM and CCM.
  • Finally, the authors should mention policy implications. There is no inclusion of articles from current evidence on school nutrition policies. I would suggest authors to referring to these articles (i.e., Children (Basel). 2018 Jan 29;5(2):18 ; PLoS One. 2018, 29,13(3), e0194555.
  • Authors should follow the journal guidelines for references.
  • Refs # 15,20,28,29,38 are very old- please update.

Reviewer 2 Report

please have a look in the attached file

Round 2

Reviewer 1 Report

No further comments.

This manuscript is a resubmission of an earlier submission. The following is a list of the peer review reports and author responses from that submission.